# Impact of Manganese on Neuronal Function: An Exploratory Multi-Omics Study on Ferroalloy Workers in Brescia, Italy

**DOI:** 10.3390/brainsci15080829

**Published:** 2025-07-31

**Authors:** Somaiyeh Azmoun, Freeman C. Lewis, Daniel Shoieb, Yan Jin, Elena Colicino, Isha Mhatre-Winters, Haiwei Gu, Hari Krishnamurthy, Jason R. Richardson, Donatella Placidi, Luca Lambertini, Roberto G. Lucchini

**Affiliations:** 1Department of Environmental Health Sciences, Robert Stempel College of Public Health and Social Work, Florida International University, Miami, FL 33199, USA; flewi017@fiu.edu; 2Department of Medical and Surgical Specialties, University of Brescia, 25123 Brescia, Italy; d.shoieb@studenti.unibs.it (D.S.); donatella.placidi@unibs.it (D.P.); 3St. Jude Children Research Hospital, Memphis, TN 38105, USA; yan.jin@stjude.org; 4Department of Environmental Medicine and Public Health, Icahn School of Medicine at Mount Sinai, New York, NY 10029, USA; elena.colicino@mssm.edu; 5Isakson Center for Neurological Disease Research and Department of Physiology and Pharmacology, College of Veterinary Medicine, University of Georgia, Athens, GA 30602, USA; imwinters@uga.edu (I.M.-W.); jason.richardson@uga.edu (J.R.R.); 6College of Health Solutions, Arizona State University, Phoenix, AZ 85004, USA; haiweigu@asu.edu; 7Vibrant Sciences LLC., San Carlos, CA 94070, USA; hari@vibrantsci.com; 8Diabetes, Obesity and Metabolism Institute, Icahn School of Medicine at Mount Sinai, New York, NY 10029, USA; luca.lambertini@mssm.edu; 9Occupational Medicine, University of Modena and Reggio Emilia, 41125 Modena, Italy

**Keywords:** manganese, metabolomics, proteomics, occupational exposure, ADRD biomarkers

## Abstract

Background: There is growing interest in the potential role of manganese (Mn) in the development of Alzheimer’s Disease and related dementias (ADRD). Methods: In this nested pilot study of a ferroalloy worker cohort, we investigated the impact of chronic occupational Mn exposure on cognitive function through β-amyloid (Aβ) deposition and multi-omics profiling. We evaluated six male Mn-exposed workers (median age 63, exposure duration 31 years) and five historical controls (median age: 60 years), all of whom had undergone brain PET scans. Exposed individuals showed significantly higher Aβ deposition in exposed individuals (*p* < 0.05). The average annual cumulative respirable Mn was 329.23 ± 516.39 µg/m^3^ (geometric mean 118.59), and plasma Mn levels were significantly elevated in the exposed group (0.704 ± 0.2 ng/mL) compared to controls (0.397 ± 0.18 in controls). Results: LC-MS/MS-based pathway analyses revealed disruptions in olfactory signaling, mitochondrial fatty acid β-oxidation, biogenic amine synthesis, transmembrane transport, and choline metabolism. Simoa analysis showed notable alterations in ADRD-related plasma biomarkers. Protein microarray revealed significant differences (*p* < 0.05) in antibodies targeting neuronal and autoimmune proteins, including Aβ (25–35), GFAP, serotonin, NOVA1, and Siglec-1/CD169. Conclusion: These findings suggest Mn exposure is associated with neurodegenerative biomarker alterations and disrupted biological pathways relevant to cognitive decline.

## 1. Introduction

Manganese (Mn) is one of the most abundant elements on earth, and an essential trace element required for critical enzyme-mediated human biological processes involved in development, growth, and brain function [1]. As an enzyme cofactor, catalyst, and gene modulator, Mn plays a key role in a wide range of physiological processes including neuronal function [2,3,4]. The Central Nervous System (CNS) is a relevant exposure target for Mn throughout early life, adulthood, and old age, with negative impacts on cognitive function [5]. Mn absorption into the body occurs through dietary intake, drinking water, and inhalation of airborne particles from industrial emissions. Occupational settings including manufacturing, mining, smelting, and welding operations are among the main sources of chronic occupational exposure to various metals including Mn [6,7]. Additionally, emerging production of pure Mn is rising to supply the demand of next-generation batteries for Electric Vehicles [8]. To date, Mn neurotoxicity has primarily been studied as a cause of Parkinsonian syndrome resulting from high occupational exposure, impaired hepatic excretion, or homozygous mutations in SLC transporter genes [9]. However, recent interest in the cognitive impacts shown in non-human primates [10], as well as in occupational and environmental studies [11] has grown. As a result, investigating the cellular and molecular processes underlying the potential mechanistic role of Mn in Alzheimer’s disease (AD) is important.

Amyloidogenesis, oxidative stress, mitochondrial dysfunction, cholinergic innervation, and neuroinflammation are hypothesized as key mechanisms in Mn-induced cognitive impacts [1,12]. It is suggested that Mn exposure contributes to AD through multiple interrelated mechanisms. Mn-induced activation of inflammatory transcription factors such as NF-κB and AP-1 leads to upregulation of proinflammatory cytokines (e.g., IL-6, TNF-α, IL-17), iNOS, and COX-2), promoting neuroinflammation and oxidative stress in the brain [13]. Concurrently, Mn disrupts mitochondrial function by impairing calcium homeostasis, inhibiting electron transport chain complexes I and II, and reducing ATP production, hallmarks of AD neurodegeneration [1]. Mn also dysregulates the cholinergic system by altering acetylcholinesterase (AChE) activity and decreasing cholinergic receptor function, both of which are implicated in cognitive decline and amyloid-β (Aβ) accumulation [14]. Moreover, Mn may enhance Aβ pathology by stimulating expression of APP and BACE1 and downregulating Aβ-degrading enzymes like IDE and neprilysin [15]. Together, these mechanisms position Mn as a potential contributor to AD pathogenesis via inflammation, mitochondrial impairment, cholinergic disruption, and Aβ dysregulation [1,12,16].

As our aging population grows, the prevalence of neurodegenerative diseases (NDs) is a major public health concern. Less than 10 percent of cases of AD and AD-Related Dementias (AD/ADRD) are familial, whereas more than 90 percent are sporadic [17]. Lifestyle, environmental factors, and gene-environment interactions are strongly suspected as AD/ADRD modifiers [18,19]. Recent studies suggest that AD/ADRD preclinical stages, identified as mild cognitive impairment (MCI), can precede disease onset for years in the absence of detectable symptoms. It is suspected that, pathological alterations gradually take place in the preclinical stage, such as increased formation of Amyloid-beta (Aβ) protein in the brain [20]. Given the brain’s control over multiple physiological processes through signaling and the release of circulating proteins and metabolites, peripheral biomarkers are valuable for the early detection of AD/ADRD risk and for guiding preventive treatments [21,22,23]. Currently, cerebrospinal fluid (CSF) is utilized to measure AD/ADRD-related neurogenic biomarkers including Aβ and phosphorylated tau protein [24]. Emerging data indicate that blood-based biomarkers may be viable surrogates to the invasive CSF measurement [25]. Metabolites such as Lauric acid, glycerophosphocholine, d-glucosaminide, beta-alanine, aspartate, asparagine, alanine, l- cysteine, l-methionine, methionine–cysteine–glutamate, l-arginine, lysine, bile acid have been reported as significantly increased in AD patients compared to controls [26,27]. Metabolomic analysis may offer more accurate diagnostic and predictive insights in individuals with MCI or in AD cases where there is a disconnect between pathological markers and cognitive function [21,26]. Therefore, metabolomic and proteomic blood-based biomarker profiling are promising tools for presymptomatic detection and clinical trial monitoring to test the efficacy of disease interventions and participants’ health.

As we previously reported in an initial exploratory nested pilot study, we enrolled a subset of Mn exposed workers along with their historical controls to assess brain Aβ deposition with Positron Emission Tomography (PET) imaging and to evaluate their cognitive function. The results of this preliminary investigation revealed significantly greater diffuse brain Aβ deposition (*p* < 0.05) in Mn-exposed workers compared to controls, with no significant differences observed in cognitive function [28]. The current follow-up exploratory study aims to investigate the metabolomic, lipidomic, proteomic (including the assessment of plasma antibodies targeting 106 autoimmune-related and a panel of 64 neuronal-associated proteins), and enriched biological pathways in plasma samples of Mn exposed ferroalloy workers, with significantly higher brain Aβ aggregation, and their historical controls. In addition, we explored plasma proteomic markers relevant to AD as potential ADRD predictors of long-term high Mn exposure. Our goal is to leverage workers with long-term Mn exposure and elevated diffuse brain Aβ aggregation, to identify significantly altered metabolites, enriched biological pathways, and hypothesized biomarkers for AD or neurogenerative disorders.

## 2. Materials and Methods

**Study design:** This pilot study targeted Mn-exposed ferroalloy workers and age- and sex-matched historical control workers from the PHIME (Public Health Impact of Metal Exposure) occupational cohort of ferroalloy workers initiated by author RL in the 1990s in the Province of Brescia, Italy [29,30,31,32,33,34]. The PHIME project enrolled also children, elderly and patients affected by neurodegenerative disorders, all residing in the same geographical area [35,36,37,38,39,40,41,42,43,44]. The initial workers cohort included more than 400 subjects, employed in ferroalloy industries located in different areas of the province of Brescia, Northern Italy: (i) three historical plants, active from 1901 to 2001 in the towns of Sellero, Darfo, Breno situated along Valcamonica, a valley of the pre-Alps region, (ii) a currently active plant in the town of Bagnolo Mella, in the plain area of the same province, Figure 1a, where ferroalloy operations started in the 1970s, with electric furnaces crushing and melting Mn ore for the production of ferro-Mn and silico-Mn alloys Figure 1b, although with progressively diminishing production due to the high cost of energy.

The concentration of airborne manganese in total dust and respirable fractions had been measured annually since the 1990s through personal and stationary sampling and analyzed using electrothermal atomic absorption spectrometry (VARIAN SPECTRAA 400 Zeeman). A cumulative exposure index (CEI) was calculated for each worker, multiplying the average annual airborne Mn concentration in total dust characteristic of each job, by the number of years in which the activity was performed, according to the following formula: CEI = {(MnA_1_ × YRS_1_) + (MnA_2_ × YRS_2_) + … + (MnA_n_ × YRS_n_)}. Therefore, CEI was expressed in µg Mn/m^3^ x years, and by dividing the total number of years (YRS_n_)_,_ it yielded an Average Annual Cumulative (AAC) concentration of airborne Mn (in µg/m^3^) for each individual worker [30,34]. For the worker group, the mean ± SD of AAC (µg/m^3^) was 1646.15 ± 2582, with a median of 528.75.

**Target population**: The study included six exposed male workers (mean age ± SD: 62.67 ± 4.4 years; median: 63) and five historical male controls (mean age ± SD: 60.80 ± 2.7 years; median: 60), who had undergone brain imaging at the University of Brescia with General Electric Discovery 690 PET-CT scanner after the injection of 185 MBq of [^18^F] flutematol (Vizamyl, GE Healthcare, Marlborough, MA, USA) through a catheter placed in an intravenous line in an antecubital vein. PET acquisition was carried out for 20 min, starting 90 min after injection. β-amyloid deposition in the ferroalloy workers resulted more diffuse than in controls (*p*, 0.05), and the major regions with increased beta-amyloid uptake were the anterior and posterior cingulate, entorhinal cortex and part of hippocampus, dorso-lateral prefrontal cortices [28].

**Sample collection:** Whole blood and plasma samples from Mn-exposed workers and controls were collected at the time of brain imaging and stored at −80 °C at the University of Brescia. Based on the participants’ consent, samples were shared with the FIU Stempel College of Public Health and Social Work in 2023, after a Material Transfer Agreement and a Data Use Agreement were co-signed by both institutions and a new IRB (IRB # 111589) was approved at FIU, USA, based on the original IRB approval from the University of Brescia.

**Manganese Assessment:** Plasma manganese concentrations were quantified using inductively coupled plasma mass spectrometry (ICP-MS) following a validated laboratory-developed protocol at Vibrant America Clinical Laboratories (Santa Clara, CA, USA). Plasma samples were collected in trace-element-free tubes and stored at 4 °C until analysis to minimize contamination and degradation. For each measurement, 400 µL of plasma was combined with 3.6 mL of ICP-MS diluent, consisting of 18 MΩ deionized water, Optima-grade nitric acid (20 mL/L), methanol (20 mL/L), Triton X-100 (0.5 mL/L), and an internal standard solution (Environmental Standard Mix 6, PerkinElmer, Waltham, MA, USA). The mixture was vortexed thoroughly to ensure complete lysis and homogeneity. Calibration was conducted using a certified single-element manganese standard (PerkinElmer TruQ™, 1000 µg/mL in 2% HNO_3_). A six-point calibration curve ranging from low to high parts-per-billion (ppb) concentrations was prepared via serial dilution, and its accuracy verified using Bio-Rad Lyphochek Urine Metals Control Levels 1 and 2. Each analytical run included standards, blanks, and quality control samples.

Plasma samples were introduced into a PerkinElmer NexION ICP-MS equipped with an ESI SC-4 DX autosampler and controlled by Syngistix software 4.0. The system was tuned daily with a certified tuning solution (PerkinElmer, Cat. No. N8145051), ensuring proper mass calibration, sensitivity, and oxide ratio performance using the SmartTune Manual workflow. Plasma was ignited and stabilized for 30 min prior to measurement, with argon gas pressure maintained at 100 psi. Sample introduction was performed using a peristaltic pump and acid-resistant tubing; 18 MΩ deionized water was used to flush the system between samples.

Manganese was detected at its most abundant isotope, ^55 Mn, and quantification was achieved through internal standard correction and external calibration. All measurements, including standards and controls, were run in duplicate, and only those meeting acceptance criteria for calibration linearity and control recovery were included in the final dataset. Analytical results were exported through Syngistix and underwent quality assurance review under the laboratory’s CLIA-certified workflow.

**Metabolomic and lipidomic Analysis:** The experimental procedures followed established methodologies in Gu’s lab for LC-MS analysis, employing specific techniques for sample preparation and data collection [41,42,43,44,45]. For metabolomics, plasma samples were initially thawed, vortexed, and treated with methanol. After incubation at −20 °C and centrifugation, supernatants were dried, reconstituted, and processed for metabolomics analysis. Lipidomic analysis involved the addition of PBS and internal standards, followed by MTBE extraction. LC-MS experiments were conducted using a Thermo UPLC-Exploris 240 Orbitrap MS (Waltham, MA, USA), with injections in both positive and negative ionization modes. A Waters XBridge BEH Amide column (150 × 2.1 mm, 2.5 μm particle size) from Waters Corporation, Milford, MA, USA, was utilized for these modes in metabolomics. In lipidomics, reverse-phase chromatography was employed using a Waters XSelect HSS T3 column. Utilizing a mass spectrometer equipped with an electrospray ionization (ESI) source, we will acquire untargeted data across 70–1000 *m*/*z* for metabolomics and 200–2000 *m*/*z* for lipidomics. Peak identification and data processing were supported by extensive databases and stringent criteria, using Thermo Compound Discoverer 3.3 and LipidSearch 4.2 software to ensure high data quality in metabolomics and lipidomics, respectively. Only peaks with a coefficient of variation under 20% across quality control pools and present in over 80% of samples were considered for further analysis. 


**Single molecule array (Simoa)**


Plasma samples stored at −80 °C were thawed and centrifuged to remove debris before analysis on the Quanterix SR-X platform using single-molecule array (Simoa) technology (Billerica, MA, USA) in Isakson Center for Neurological Diseases Research, University of Georgia. The Simoa Neuro 3-Plex A (N3PA) Advantage Kit (Catalog No. 101995) used to measure Aβ40, Aβ42, and Total Tau, while the Simoa Neuro 2-Plex B (N2PB) Advantage Kit (catalog no. 103520) used for quantitative determination of GFAP (Glial Fibrillary Acidic Protein) and neurofilament light (NfL) biomarkers. Duplicates of all standards and targets were run on a 96-well plate, with detection based on fluorescent signal formed by an immunocomplex consisting of bead, bound protein and detection antibody. Signals were measured as Average Enzymes per Bead (AEB), and concentrations were interpolated using a standard curve with a weighted 4-parameter logistic fit.


**Protein microarray (protein chip)**


A protein microarray (protein chip) was used to monitor and analyze changes in 106 autoimmune-related and a panel of 64 neurological conditions-associated (Neuronal Zoomer) antibodies in plasma samples from both manganese-exposed workers and control group, conducted by Vibrant America LLC in Santa Clara, CA, USA. The antigens corresponding to these antibodies were fixed onto silicon wafers at a concentration of 1.0 μg/mL and incubated for 24 h at 4 °C. Unbound antigens were removed, and the wafers were neutralized with a blocking solution. The wafers were then cut into microchips, each containing a specific antigen, and these were mounted onto a 24-pillar plate with 44 microchips per pillar.

For the protein chip assay, plasma samples were diluted 1:20 and incubated on the pillar plate for 1 h at room temperature. This was followed by washing steps and incubation with secondary antibodies (Goat Anti-Human IgG HRP and IgM HRP) at 1:2000 dilution. After further washing and air drying, a chemiluminescent substrate was added. The plates were scanned for 5 min using a high-resolution chemiluminescence imager (Q-View™ Imager Pro, Quansys Biosciences) [45]. 

**Statistical Analysis:** Initial analyses involved descriptive statistics and Fisher’s Exact Test for categorical variables. Student’s *t*-test was used to compare the mean values between the two groups of exposed and control workers. Given the potential for skewed distributions in metabolite and lipid concentrations, log transformation was applied to meet the assumptions of normality and homogeneity of variances. ANCOVA (Analysis of Covariance) was utilized to adjust for confounders such as age, providing a more accurate assessment of the impact of manganese exposure on the outcomes. PCA (Principal Component Analysis) and PLS-DA (Partial Least-Squares Discriminant Analysis) were employed to visualize the complex data from metabolomic studies into fewer dimensions, which facilitated the identification of patterns, groupings, and outliers [46].

Pathway and Metabolites Enrichment Set Analyses were conducted using MetaboAnalyst 6.0 web-based platform to elucidate the biological significance of metabolomic or lipidomic data. The pathway analysis was conducted using the table of metabolite concentrations or compound listed in over-representation analysis (ORA), with the organism-specific pathway sets of Homo sapiens KEGG (Kyoto Encyclopedia of Genes and Genomes) database used as the reference background. The global test was used to assess the significance of groups of metabolite data points and determine whether a set of features was collectively associated with a certain biological pathway more than would be expected by chance. In addition, the analysis incorporated a network topological metric known as relative-betweenness centrality. This measure evaluates the extent to which a node (a given metabolite in this context) acts as a bridge or connector within the network paths. Relative-betweenness centrality provides insight into the importance of a node in terms of information flow or interaction within the network structure, normalized to a scale that allows for comparison across different networks or nodes. In conducting Metabolites Enrichment Set Analysis (MESA), the log 10 transformed normalized metabolite data was analyzed using the Rapid Metabolite Profiling Small Database (RaMP-DB), a reference of 3694 metabolite and lipid pathways (integrating KEGG via HMDB, Reactome, WikiPathways) in Homo sapiens. By combining these methods, the analysis aimed to elucidate the roles of specific metabolites in the context of their network positions and their collective relevance to a biological condition or phenotype [47,48]. All metabolomics, lipidomics, and proteomics analyses were performed in a single batch per sample type. Each assay included technical replicates of the same samples for quality control and consistency.

The linear regression model was used for biomarker prediction to predict the value of the dependent variables (Metabolite Concentrations, Protein levels) based on the value of the independent variable (Plasma Mn).

Statistical significance was assessed using a combination of methods: unpaired two-tailed Welch’s *t*-tests for unadjusted group comparisons, and ANCOVA or linear regression models to adjust for covariates such as age and sex. For multiple comparisons, *p*-values were adjusted using the Benjamini–Hochberg false discovery rate (FDR) method. Significance was defined as *p* < 0.05 or FDR-adjusted *p* < 0.05, with an additional threshold of fold change (FC) > 1.5 applied where relevant.

The analyses were conducted using RStudio (v4.1.0), Python (v3.11.4), SPSS 29.0.1.1, Excel, MetaboAnalyst 6.0, and LIPEA 1.5.0 (Lipid Pathway Enrichment Analysis) bioinformatics tool. The study acknowledges limitations such as the small sample size and the consequent potential biases.

## 3. Results


**Descriptive characteristics of the target groups**


The characteristics of Mn-exposed workers and historical unexposed controls were analyzed using the Chi-square test and the summary statistics are reported in Table 1.

Age was tested as a categorized variable with two ranges 54–61, and 62–69. The corresponding *p*-values of Phi value-Cramer’s V (coefficient data) from all 2 × 2 contingency tables presented in the heatmap, Figure 2. No significant differences were observed for age, smoking, alcohol consumption, family history of neurodegenerative diseases, and comorbidity (such as hypertension and cardiovascular diseases) as shown in Figure 2.

As a potential risk factor for ADRD, age at time of sampling was considered in the statistical analysis including ANCOVA and multivariate regression, regardless of non-significant correlation between exposure and age.


**Exposure data**


Plasma manganese (PMn) levels (ng/mL), measured using Inductively Coupled Plasma Mass Spectrometry (ICP-MS), differed significantly between exposed workers and controls (*p* = 0.027). The exposed group showed higher concentrations (mean ± SD: 0.704 ± 0.20; median: 0.67) compared to controls (mean ± SD: 0.397 ± 0.18; median: 0.45), supporting evidence of occupational Mn exposure as shown in Figure 3.


**Metabolomic analysis**


After normalization through log 10 transformation, 1317 metabolites with a CV (QC) less than 20 percent were included in the multivariate statistical techniques including PCA and PLS-DA for exploratory data analysis and predictive modeling. Principal Component Analysis (PCA) of the metabolomic data reveals a separation trend with partial overlap between Mn-exposed and control groups, while Partial Least Squares Discriminant Analysis (PLS-DA) demonstrates a clearer distinction between the two groups as shown in Figure 4.

The nonage-adjusted Student’s *t*-test comparing the two groups revealed statistically and biologically significant differences (*p* < 0.05, FC > 1.5), including an increase in 49 and a decrease in 13 metabolites in the Mn-exposed compared to controls (Appendix A). After adjusting for age ANCOVA identified 98 significantly different metabolites between two groups (*p* < 0.05, FC > 1.5), Figure 5.

In the pursuit of understanding the impact of manganese exposure on metabolic processes, we engage in a detailed investigation and in-depth analysis using linear regression. Within this framework, Plasma manganese level was treated as the independent variable, metabolites concentrations as the dependent variable, and age as a covariate. The linear regression model identified 203 metabolites significantly associated with plasma Mn, Figure 6.

The pathway analysis has revealed that amino acid metabolism pathways, particularly those involving the biosynthesis of tyrosine, phenylalanine, and tryptophan, as well as the metabolism of alanine, aspartate, and glutamate, were significantly affected among the exposed workers. These pathways were consistently identified across non-age adjusted *t*-test (Appendix A), age-adjusted ANCOVA, Figure 7, and linear regression analyses, Figure 8, as primary targets of impact.

Metabolite Set Enrichment Analysis (MSEA) in Mn-exposed and control workers

The metabolomic enrichment analysis in non-age adjusted *t*-test displayed olfactory signaling pathway, and mitochondrial fatty acid beta-oxidation among the top enriched metabolite set list (Appendix A). The metabolites in the impacted set that are significantly changed (Hit) in the experimental data are shown in Appendix A. ANCOVA was conducted to adjust the metabolomic findings, accounting for age as potential confounding variable and main risk factor for ADRD. The metabolite enrichment analysis, leveraging distinct metabolites identified by ANCOVA, indicated that biogenic amine synthesis, transmission across chemical synapses, neuronal system, 22q11.2 copy number variation syndrome and neurotransmitter clearance were notably affected in Mn-exposed workers compared to control, Figure 9.

Linear regression was utilized for a detailed examination of how the levels of plasma Mn influence and predict the metabolite profiles, with age factored in as covariate. This analysis identified a significant correlation between plasma Mn and various metabolic processes in the enrichment study, including SLC-mediated transmembrane transport, transport of small molecules, biochemical pathways, transport of bile salts and organic acids, metal ions and amine compounds, and transport of inorganic cations/anions, Figure 10.


**Lipidomic analysis**


After normalization through log 10 transformation, 189 lipids with a CV (QC) less than 20 percent were included in the subsequent statistical analysis including PCA, PLS-DA (Figure 11), ANCOVA, Linear regression, pathway analysis, and enrichment analysis. Following exploratory analysis, ANCOVA identified three lipids- SM (t36:2), SM (t38:2), and MePC (26:3e)- as significantly different between the exposed and control groups; however, the effect size was not significant.

The Linear regression model identified associated lipids with PMn including Sphingomyelins (SM), Phosphatidyl choline (PC), Lysophosphatidylcholine (LPC), and lyso dimethyl phosphatidylethanolamine (LdMePE) as significantly increased in the exposed workers compared to the controls. Specifically, lipids, SM (d41:4), PC (38:1), PC (36:5e), PC (38:4e), LPC (18:1), LPC (18:2) and LdMePE (18:1) were increased in the exposed workers compared to the controls (Figure 12). Enriched lipid pathway analysis using LIPEA revealed Glycerophospholipid metabolism, Choline metabolism in cancer, Necroptosis and Retrograde endocannabinoid signaling as key impacted pathways in the exposed group, Table 2.


**Proteomics and AD Plasma Biomarkers**


Using the ultrasensitive Simoa platform to evaluate ADRD-related protein biomarkers showed increased yet statistically non-significant levels of Aβ42, Aβ40, total tau, NfL, and GFAP in the plasma of manganese-exposed workers, alongside a reduced Aβ42/Aβ40 ratio, compared to the control subjects, Figure 13 and Appendix A.

Proteomic profiling using assessing antibodies (immunosignaturing) targeting 106 autoimmune-related, Appendix A, and 64 neurological conditions associated (Neuronal Zoomer), Appendix A, proteins, performed using the very sensitive high-throughput protein microarray technology, identified 13 proteins to be differentially expressed in the plasma of workers. Except for Anti Mitochondria antibodies M2, which were decreased, all other differentially expressed antibodies including GM2, Beta-Amyloid (25–35), Tubulin Beta (TUBb), 5-Hydroxytryptamine (5-HT, Serotonin), Human DBH/Dopamine Beta-Hydroxylase Protein, Human NOVA1 Protein, Human Siglec-1/CD169 Protein, Recombinant Adaptor Related-Protein Complex 3 Beta 2 (AP3b2), Human Brain Cerebellum Protein, Recombinant Human GFAP Protein, Recombinant Human Amphiphysin/AMPH Protein, and rhCarbonic Anhydrase VIII were found to be upregulated in the plasma of Mn-exposed workers compared to control group significantly (*p* < 0.05) as indicated in Figure 14 and Appendix A.

## 4. Discussion

This pilot study explores the mechanisms linking chronic Mn exposure to neuronal dysfunction. The central hypothesis of this study is that chronic exposure to Mn is associated with neurodegenerative pathology contributing to amyloidogenesis, which is linked to cognitive impairment and biomolecular alterations. This analysis identified several key metabolic and lipidomic pathways that are prominently featured in this cohort, including olfactory signaling, mitochondrial fatty acid beta-oxidation, biogenic amine synthesis, SLC-mediated transmembrane transport, necroptosis, glycerophospholipid metabolism, and choline metabolism. The findings of proteomic profiling revealed elevated tau protein levels and a reduced Aβ42/Aβ40 ratio in Mn-exposed individuals. Additionally, significant alterations were identified in 11 neurologically relevant antibodies, including Aβ25–35, GFAP, NOVA1, AMPH, Serotonin, GM2, Tubulin Beta (TUBb), 5-HT, Dopamine Beta-Hydroxylase, AP3b2, Human Brain Cerebellum Protein and two autoimmune-related proteins CD169 and Anti-Mitochondria Antibodies M2. These results highlight potential neurological impacts of Mn exposure.


*
Metabolomics profiles among exposed and control workers
*


Metabolomics aids the identification of new biomarkers by providing a dynamic and detailed snapshot of the metabolic status of organisms. In this study, we aimed to describe the metabolomic profile in the plasma of ferroalloy workers and individuals with no metal exposure to distinguish the patterns between two groups. The metabolites circulating in the bloodstream can have implications for brain health by either impacting or reflecting the metabolic processes of the brain. These metabolites might interact with or pass through the blood–brain barrier, thereby affecting or reflecting brain metabolism. Research on metabolites associated with brain health has largely been performed in the context of neurodegenerative diseases and cognitive functions. However, fewer studies have extended to include neuroimaging indicators related to these conditions that could be induced by metal exposure [49].

In this study, the metabolomic enrichment analysis in non-age adjusted *t*-test displayed olfactory signaling pathway and mitochondrial fatty acid beta-oxidation in top enriched metabolite set list (Appendix A). The key metabolites identified were butyric acid, propionic acid, geranyl acetate, choline, phenylalanine, and glutamic acid. The PHIME cohort study includes an occupational cohort that generated this pilot study, and has shown olfactory impacts of Mn exposure among the workers [31], but also among children [50] and elderly [41] living in the same region of the workers and environmentally exposed to the emission from the ferroalloy industry [42]. Mn-related olfactory dysfunction is recognized as a non-motor dysfunction and early predictive sign in various human and animal models of neurodegenerative diseases. Different mechanism contributed to Mn-induced olfactory alteration including Mn-accumulation in the olfactory bulb, neuroinflammation, dopaminergic dysregulation, loss of Gamma-aminobutyric acid (GABA)-ergic interneuron, impaired neurogenesis through decreases the survival of adult-born cells and substantially hinders [51,52,53,54,55].

The current study has revealed significant alterations in mitochondrial pathways, notably the beta-oxidation of fatty acids and the l acid (TCA) cycle, which are essential components of cellular energy metabolism. These pathways were among the most prominently affected metabolite sets identified. Mitochondria function in neuronal cells is crucial due to substantial demand for energy and restricted glycolytic ability in these cells. Mitochondrial dysfunction is associated with neurotoxicity triggered by Mn exposure, which includes amyloid and tau pathology, and is also identified as an initial indicator of Alzheimer’s disease [56,57]. Energy metabolism, together with additional factors such as oxidative stress and action of mitochondrial antioxidant enzyme manganese superoxide dismutase (MnSOD); which is the primary ROS scavenging enzyme in the cell, mtDNA damage, calcium mishandling, alteration in the morphology, dynamics, and axonal transport of mitochondria, and inflammation all of which play a role in Mn induced mitochondrial dysfunction [58,59,60,61].

Metabolite enrichment analysis, using distinct metabolites identified by ANCOVA, revealed that biogenic amine synthesis and transmission across chemical synapses are significantly impacted in Mn-exposed workers compared to the control group. Choline, Glutamic acid, Phenylalanine, Homovanillic acid, 3,4-Dihydroxybenzeneacetic acid were the key metabolites predominantly impacted in these sets (Appendix A).

Biogenic amines encompass critical neurotransmitters such as dopamine, norepinephrine, epinephrine, serotonin, and histamine, essential to neural function. Amino acids undergo various modifications including amine acid decarboxylation, hydroxylation, methylation, transamination and beta-oxidation to form these neurotransmitters. The synthesis of biogenic amines is tightly regulated in the body since their dysregulation is linked to various neurological and psychiatric disorders. Alterations in biogenic amines and amino acids have been linked to AD and dementia, as evidenced in numerous studies. On the other hand, the pathological accumulation of Aβ [62,63,64,65,66]. The effect of Mn on biogenic amine in various regions of the brain was documented in different human and animal studies. Mn accumulation in certain regions of the brain may alter neurotransmitter concentration or activity including dopamine, choline, glutamate, and γ-aminobutyric acid [63,67,68,69,70,71].

Linear regression showed an association between plasma Mn, and the two top enriched metabolite sets SLC-mediated transmembrane transport, and the movement of small molecules. The amino acids and related small molecules with notable associations in these metabolite sets are L-Carnitine, Citric acid, Hypoxanthine, L-Tyrosine, Proline, Lactic acid, Uracil, Creatinine, N-Acetylneuraminic acid, L-Aspartic acid, L-Valine, L-Threonine, 4-Hydroxyproline (Appendix A). The SLC-mediated transmembrane transport emerged as the most significantly enriched metabolite set in this analysis. This pathway encompasses a major superfamily of membrane transporters that facilitate the transportation of ions, signaling molecules, organic and bile acids, and nutrients. Within the brain, SLCs transporters are pivotal in energy metabolism (i.e., SLC2A, the main glucose transporter), the glutamate/GABA-glutamine cycle (i.e., vesicular transporters of SLC17A6/A7 and SLC32A1), neurotransmitter release and reuptake (i.e., SLC22A5, in the BBB which modulates carnitine levels, potentially enhancing acetylcholine synthesis, reducing oxidative stress, and prevent neurodegeneration [72].

Studies on ADRD have documented various mechanisms involving SLC transporters: diminished brain glucose metabolism linked to SLC2A1 and SLC2A2, Aβ deposition associated with reduced SLC2A1 expression, tau hyperphosphorylation related to loss of neuron-specific SLC2A3. Additionally, an increase in SLC2A2 has been observed due astrocyte activation in AD, while a reduction in SLC1A2, particularly within the hippocampal and prefrontal cortex regions, has been noted in the advanced stages of AD and during the progression of mild cognitive impairment [73,74,75,76,77,78,79,80].

Moreover, the importance of SLCs in Mn transport has been highlighted in numerous studies. Mutations in SLC30A10, SLC39A8 and SLC11A2 (DMT1) have been linked to alteration in Mn homeostasis, leading to neuronal dysfunction [81,82,83,84]. The enrichment of SLC-mediated transmembrane transport pathways could represent a compensatory response to elevated levels of blood and plasma Mn in workers, serving as a protective mechanism for the body.


*
Lipidomic analysis
*


Enriched lipid pathway analysis showed significantly different lipids, and linear regression revealed necroptosis, sphingolipid signaling pathway, glycerophospholipid metabolism, and choline metabolism, as key impacted pathways. These findings corroborate multiple studies that have linked increased plasma level of choline and phosphatidylcholine (PC) with early-stage pathological changes, such as Aβ deposition, and with later-stage clinical manifestations, including cognitive dysfunction [5,14,85,86]. Structural modification and reduction in specific brain phosphatidylcholine (PC) and phosphatidylethanolamine, and Sphingomyelin have been reported among AD patients and MCI cases from the Alzheimer’s Disease Neuroimaging Initiative (ADNI) database and proposed as potential biomarkers in AD [87,88,89,90].

Alzheimer’s disease and other neurodegenerative disorders are marked by sever neuronal loss, with reports indicating the activation of necroptosis in the brains. This finding is aligned with previous reports identifying activated necroptotic markers form insoluble amyloid-like structures, and their colocalization mainly within neurons, rather than other cell types, in postmortem AD human brains. Additionally, a negative correlation between the cognitive performance, as measured by mini-mental state examination (MMSE), and the degree of necroptosis has been detected [91,92,93].

Glycerophospholipid (GP) and sphingolipid metabolism are two critical enriched lipid pathways not only involved in maintaining the structural integrity of neuronal membranes. These lipids along with cholesterol are integral components of the signaling pathways. Prior studies reported substantial alterations in the level of these lipids in the neocortex of postmortem brain in AD cases compared to age-matched control [94]. It has been observed that reduced concentrations of GPs (such as phosphatidylinositol (PI), phosphatidylethanolamine (PE), and phosphatidylcholine (PC)) in brain correlate with an increased formation of neurofibrillary tangles and amyloid pathologies, alongside a less efficient clearance of amyloid-beta (Aβ). This depletion alters the membranes’ permeability and fluidity, disrupts ion homeostasis, and escalates oxidative stress. Degradation byproducts of GPs, known to provoke inflammation, activate microglia and astrocytes, triggering the release of inflammatory cytokines that further contribute to Aβ accumulation. Moreover, from our earlier discussion, the affected beta-oxidation of fatty acids pathway in mitochondria, might be due to Mn-induced oxidative stress leading lipid peroxidation [86,95,96].

The cholinergic system is implicated in manganese (Mn) toxicity, potentially impacting choline uptake due to a possible computation with choline transporters in BBB, quantal release of acetylcholine into the synaptic cleft, synaptic degradation, alteration in AChE function which could affect amyloid beta fibril formation, and binding acetylcholine with proteins in astrocytes [1,97,98].


*
Proteomics
*


Blood-based biomarkers of ADRD are emerging as promising predictors due to rapid advances and development of highly sensitive tests for detecting brain pathology [99]. Such pathologies can begin to develop well before symptoms manifest. Our novel study investigated pre-symptomatic, brain-related biomarkers, such as blood -based proteins, in Mn-exposed ferroalloy workers, which exhibited significantly higher level of diffuse Aβ in different parts of brain [28].

Emerging research shows that high manganese concentrations can influence plasma amyloid-beta and tau levels, posing a potential risk for Alzheimer’s disease [100,101]. In our study, the ultrasensitive Simoa platform has shown increased, albeit statistically non-significant, levels of Aβ42, Aβ40, total tau, NfL, and GFAP in the exposed workers to manganese, along with a decreased Aβ42/Aβ40 ratio when compared to control subjects. Inconsistencies are present in the literature regarding changes in Aβ40 and Aβ42 levels, with both elevations and reductions reported in conditions of MCI and AD. Aβ42, being a predominant C-terminal variant present in brain Aβ plaques, and the decreased Aβ42/Aβ40 in blood have been confirmed as one of the validated blood biomarkers inversely related to brain plaque burden, as measured by PET scans [102]. Total tau, and specifically phosphorylated tau (p-tau), constitute a second, more consistent blood biomarker for AD, exhibiting increased levels in cases with MCI and AD. A comprehensive study involving the ADNI and BioFINDER cohorts has verified elevated plasma tau concentrations in patients with Alzheimer’s disease dementia and marked correlations between levels of plasma tau and subsequent cognitive deterioration [103]. Aβ42/Aβ40 and APP669-711/Aβ42 are recognized as biomarkers indicative of brain amyloidosis, whereas tau signify acute neuronal injury and is not specific biomarkers of Alzheimer’s disease [24]. The findings of our study align with prior research, substantiating an elevated level of total tau (threefold higher than controls) and a diminished Aβ42/Aβ40 ratio (approximately half that of controls) in the Mn-exposed group, consistent with the diffuse Aβ deposition observed at the PET scan. The levels of NfL and GFAP have shown a moderate increase in the worker group, as is typically seen in AD. The small sample size may account for the absence of statistically significant results, failing to reach the *p*-value threshold of 0.05.


*
Immunosignaturing
*


The role of the immune system in the development of Alzheimer’s disease has been demonstrated through various mechanisms. These include inflammation triggered by Aβ, along with increased level of inflammatory factors such as IL-10, Tfh, TNF-***α***, IL-1β in activated astrocyte and microglial cells which contribute to Aβ aggregation. In response to AD, microglial cells become phagocytic and release various inflammatory mediators, including cytokines, chemokines, growth factors, complement molecules and adhesion molecules [104,105]. The other mechanism involves the association of complement proteins with affected tissues such as senile plaques, leptomeninges and congophilic vessels, correlating with the severity of AD [106,107,108]. Additionally, studies have found an association between elevated cytokines markers associated with APOE-ε4 allele, a higher risk allele for AD [109]. There are also reports on the Mn on neuroinflammatory mediators and inflammation-induced amylogenesis [110].

Immunosignaturing and antibody profiling play an important role in identifying diseases and monitoring immune responses, which may contribute to neuronal pathology. AD-related plasma autoantibodies are emerging as potential candidate biomarkers offering diagnostic value in preclinical and prodromal stages [111,112,113]. Different anti-Aβ monoclonal antibodies (mAbs), such as Aducanumab and Lecanemab are being explored in clinical trials for their immune therapeutic potential against AD, although their effectiveness remains to be improved [114].

The assessment of plasma antibodies targeting 106 autoimmune-related and a panel of 64 neuronal-associated proteins (Appendix A), identified 13 antibodies with differential expression in the plasma of exposed workers. Except for anti-mitochondrial antibodies M2, which were decreased, all differentially expressed antibodies were found to be elevated significantly in the exposed compared to the control group (Figure 14, Appendix A), including a relevant increase in antibody targeting the protein amyloid beta 25–35, a biologically active fragment of the full-length Aβ peptide. An elevated level of this antibody could contribute to immune response by reducing Aβ levels. However, there are no consistent reports regarding the level of Aβ antibody in AD patients compared to MCI cases and controls. Lie et al. showed that the levels of naturally occurring Abs-Aβ (NAbs-Aβ) targeting the amino terminus of Aβ increased, whereas those targeting the mid-domain of Aβ decreased in both CSF and plasma in cases with MCI and AD. A decrease in Aβ antibody has also been reported in AD patients. Further research is needed to validate the implication of these findings [115,116,117].

Another protein exhibiting an elevated level of targeted antibody response is GFAP. It has been shown that GFAP level in individuals without dementia correlates with amyloid burden, and those who are amyloid-positive are linked with neuroimaging biomarkers. An increased Recombinant human GFAP has been reported as a biomarker for AD-related pathogenesis prior to dementia [118,119].

A significantly increased Tubulin beta (TUBb) antibody, observed in this study, could contribute to alterations in microtubule stability and tau protein interactions, which are hallmarks in the progression of Alzheimer’s [120].

Gangliosides like GM2, which are glycosphingolipids in the central nervous system, have been found to accumulate in the brains of those with neurodegenerative diseases. There is substantial evidence supporting the clinical relevance of cerebrospinal fluid glycosphingolipid measurements and the presence of anti-glycosphingolipid antibodies, although further research is needed to elucidate their implications [121].

Serotonin (5-Hydroxytryptamine, 5-HT) regulates mood and cognitive functions. Alterations in serotonin levels and serotonergic pathways have been observed in Alzheimer’s disease, impacting mood and cognitive abilities. Studies have shown increased levels of serotonin and other neurotransmitters like dopamine in cases with aggregated amyloid-beta and neurodegenerative diseases. The enzyme dopamine beta-hydroxylase (DBH), which converts dopamine to norepinephrine, is also significant in neurotransmitter regulation. Imbalance in these neurotransmitter systems has been implicated in various neurological conditions including neuroinflammation which is prevalent in Alzheimer’s disease [122,123,124].

The Human NOVA1 Protein, crucial in neuronal development, is associated with neuron-specific alternative splicing. It acts as a master regulator in splicing of genes involved in synapse formation. Recent studies have highlighted the role of alternative splicing (AS) in gene expression reprogramming linked to functional changes observed in Alzheimer’s disease (AD) patients [125,126].

The Human Siglec-1/CD169 Protein, a type of sialic acid-binding immunoglobulin-type lectin, primarily serves as a sialic acid receptor on macrophages located in secondary lymphoid organs. Additionally, CD169-expressing macrophages are found within the central nervous system (CNS), specifically in the meninges and choroid plexus, where they play key roles in neuroinflammation—a critical aspect of Alzheimer’s disease pathology. Furthermore, inhibitory Siglec signaling proteins modulate proinflammatory immune responses and phagocytosis in microglia, contributing to the regulation of immune responses in the brain, which is integral to the neuroinflammatory component of Alzheimer’s pathology [127,128].

The Recombinant Adaptor-Related Protein Complex 3 Beta 2 (AP3B2) plays a crucial role in directing membrane proteins to lysosomes, lysosome-related organelles, and determining the fate of synaptic vesicles, disruptions of which can impact neuronal function. Autoantibodies against the AP3B2 subunit have been linked to diseases that affect the cerebellum and vestibulocerebellum [129,130].

Anti-mitochondrial antibodies M2 (AMA-M2) are primarily formed against mitochondria, mainly in liver cells, and are associated with primary biliary cirrhosis (PBC). The specific targets and the role of these antibodies in neurodegenerative diseases have not yet been established. However, there may be a connection between manganese and AMA-M2 through the impact on mitochondrial health and function, as well as the role of bile in manganese excretion [131]. While mitochondrial damage is typically expected to elevate anti-mitochondrial antibody (AMA-M2) levels, we observed a decrease in AMA-M2 among manganese-exposed workers. One possible explanation is that chronic manganese exposure induces immune suppression or tolerance, leading to reduced autoantibody production despite mitochondrial stress. Additionally, AMA-M2 is most associated with autoimmune liver injury (e.g., primary biliary cholangitis), and its role in neurotoxicity or systemic mitochondrial dysfunction remains unclear. Given that manganese disrupts both mitochondrial function and hepatobiliary excretion pathways, the observed decrease may reflect impaired antigen presentation or altered immune response dynamics rather than the classical injury-driven elevation seen in autoimmune conditions [7,84,132,133].

The Human Amphiphysin/AMPH Protein is a brain enriched protein particularly significant in synaptic vesicle endocytosis. Elevated levels of cerebrospinal fluid (CSF) AMPH have been documented in some studies of patients with Alzheimer’s Disease (AD) and mild cognitive impairment (MCI) who exhibit abnormal tau levels. Conversely, reduced AMPH levels have been observed in brain regions with phosphorylated tau (p-tau) accumulation [134,135].

Carbonic Anhydrase enzymes are involved in pH regulation and carbon dioxide removal. Studies in animal models reported their role in neuronal signal transmission and cognitive and memory function through mechanisms of GABA-mediated synaptic transformation and ERK-related transcription. CA VIII function in neuronal calcium metabolism and nerve growth factor signaling and motor control has been reported in animal models [136,137,138,139].

Protein-Metabolites interaction analysis using a linear regression model designating proteins as the independent variable and metabolites as dependent variable adjusted for age showed that the common associated metabolites with both GM2 and Amyloid Beta (25–35) proteins, shared more metabolites with significantly different metabolites between two groups have been identified earlier. This could be due to the more potent relationship between these metabolites with mechanisms which these proteins involved in Mn-induced neurotoxicity, Figure 15.

Together, these metabolomic and proteomic findings support the role of manganese in disrupting key biological mechanisms that may promote amyloid-beta aggregation and elevate the risk of Alzheimer’s disease. Manganese toxicity may associate with its role as a specific cofactor for enzymes such as manganese superoxide dismutase (critical for oxidative stress response), arginase (involved in urea production and ammonia detoxification in the liver), and pyruvate carboxylase (essential for gluconeogenesis, lipogenesis, and the citric acid cycle). Mn also serves as a nonspecific cofactor for many other enzymes and inhibits enzymes that compete for similar binding sites, such as iron-dependent enzymes in mitochondria, leading to severe energy deficiency.

Building on these preliminary findings, future studies should aim to expand the cohort size to validate the observed associations between chronic manganese exposure and Alzheimer’s disease-related biomarkers. Longitudinal designs would help clarify temporal relationships and causality, particularly with respect to beta-amyloid accumulation and changes in metabolic, lipidomic, and proteomic profiles. Incorporating neuroimaging alongside multi-omics approaches in a time-resolved manner could further elucidate the molecular pathways linking Mn exposure to neurodegeneration. Additionally, mechanistic studies—using in vitro and in vivo models—are warranted to investigate how Mn alters amyloid processing, mitochondrial function, and neuroinflammation. Finally, research exploring gene-environment interactions (e.g., APOE genotype) may uncover differential susceptibility to Mn-induced neurotoxicity, ultimately guiding personalized occupational health interventions and biomarker-informed surveillance programs.

## 5. Conclusions

The purpose of this pilot study was to investigate the impact of chronic occupational manganese exposure on beta-amyloid deposition and biomolecular changes in the context of predisposition for Alzheimer’s disease risk among ferroalloy workers in Brescia, Italy.

The findings provide a unique foundation for further exploration of the impact of occupational manganese exposure on a range of neurological biomarkers. This nested pilot study of Mn-exposed workers integrates imaging data—previously reported to show significantly higher diffuse brain Aβ aggregation in exposed individuals—with newly characterized alterations in proteins, plasma autoantibodies targeting autoimmune and neuronal-associated proteins, and metabolic and lipidomic profiles linked to neurological conditions. Neuronal markers, whether newly identified in this study or previously established, may serve as potential biomarkers of manganese-induced amyloidogenesis. It identified several key metabolic and lipidomic pathways that are prominently featured in this cohort. These pathways are linked to manganese exposure and neurodegenerative indicators, notably amyloid beta pathology, indicating a potential link between chronic manganese exposure and biomarkers of neurotoxicity Accordingly, this study emphasizes that efforts to mitigate exposure through more stringent workplace regulations, monitoring, and the use of personal protective equipment are crucial for safeguarding worker health and reducing the burden of manganese-related occupational illnesses on public health systems. The significant results provide valuable insights that could shape the design of future studies, ideally involving a larger cohort of participants and deepening understanding of Mn-induced neurotoxicity.

## Figures and Tables

**Figure 1 brainsci-15-00829-f001:**
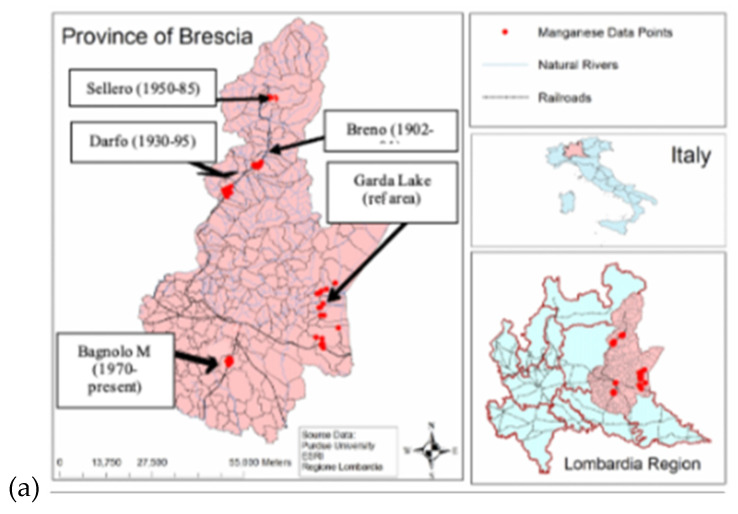
(**a**) The industry plants of Sellero, Darfo, Breno, Bagnolo Mella, and the reference area of Garda Lake in the province of Brescia, northern Italy. (**b**) Workplace areas including furnaces and casting.

**Figure 2 brainsci-15-00829-f002:**
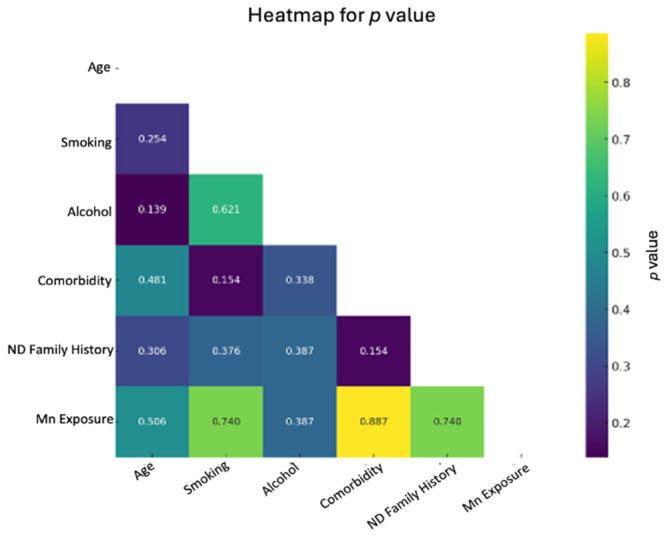
Heatmaps for *p* values. The heatmap, calculated using Fisher’s Exact Test, visually represents the strength and direction of correlation between variables, indicating the association’s pattern. The heatmap indicates corresponding *p*-values of Phi value-Cramer’s V (coefficient data).

**Figure 3 brainsci-15-00829-f003:**
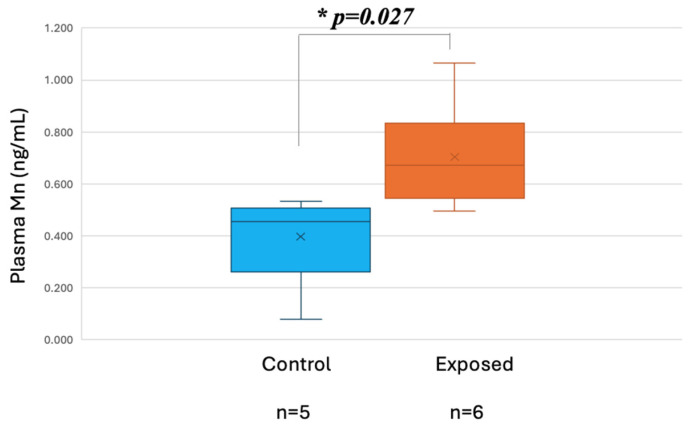
Plasma Mn (PMn) in exposed and controls. Asterisk (*) denotes statistically significant differences with *p* < 0.05.

**Figure 4 brainsci-15-00829-f004:**
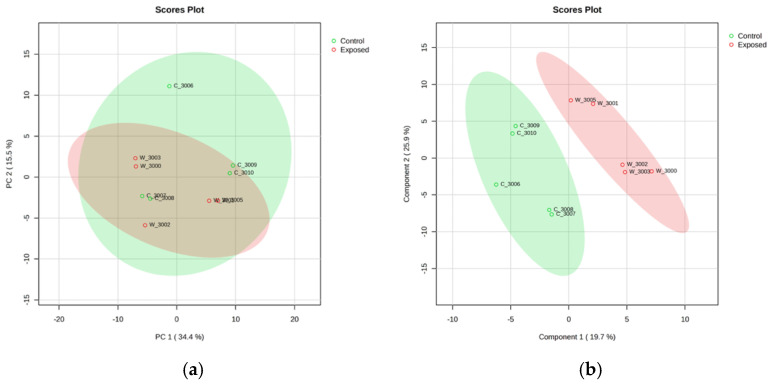
(**a**) Principal Component Analysis (PCA) of metabolomic data. It indicates a pattern of separation with some areas of distinction and some overlap between Mn-exposed (Red) and control (Green) groups. (**b**) Partial Least Squares Discriminant Analysis (PLS-DA) of metabolomic data. It shows a distinct discrimination between Mn-exposed and control groups.

**Figure 5 brainsci-15-00829-f005:**
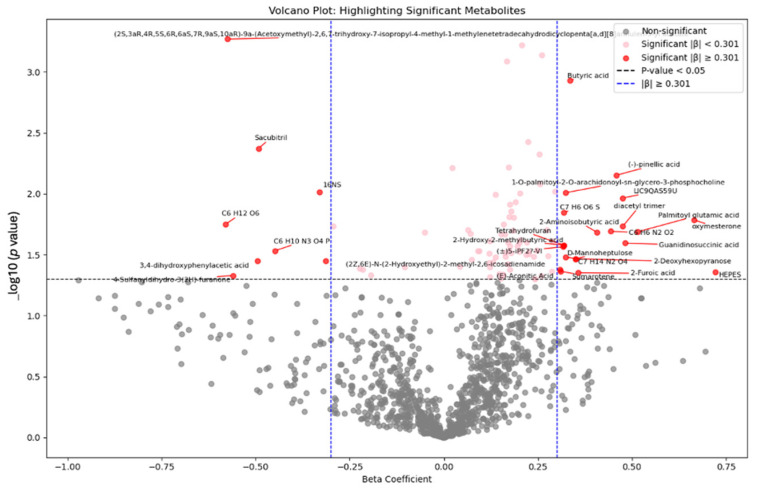
The volcano plot depicted the ANCOVA results comparing metabolites between the manganese-exposed and control groups adjusting for age. Each point represents a metabolite, with the X-axis indicating the beta coefficient (the strength and directionality of change in metabolite level), and the Y-axis showing the *p* value. The thresholds for beta coefficient representing a 2-fold change (β = 0.301), and *p* < 0.05.

**Figure 6 brainsci-15-00829-f006:**
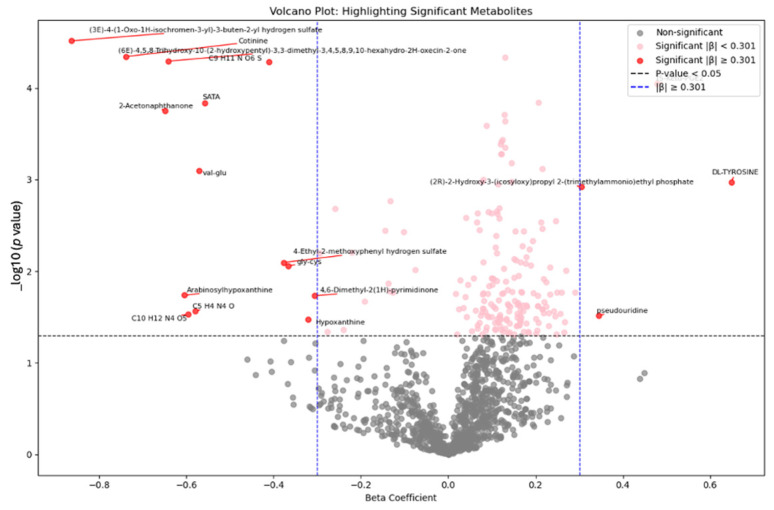
Volcano plot depicted the linear regression analysis results comparing metabolites as a dependent variable, plasma manganese as independent variable, and age as covariate. Each point represents a metabolite, with the X-axis indicating the beta coefficient (the strength and directionality of change in metabolite level), and the Y-axis showing the *p* value. The thresholds for beta coefficient representing a 2-fold change (β = 0.301), and *p* < 0.05.

**Figure 7 brainsci-15-00829-f007:**
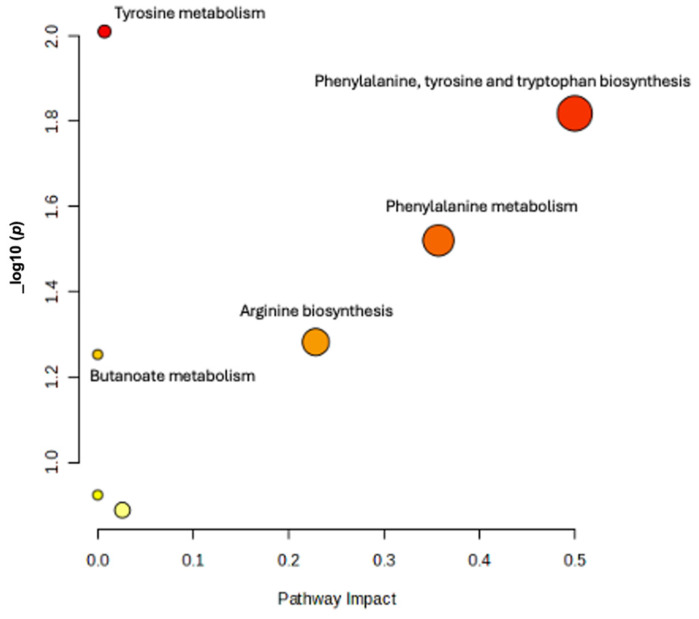
Scatter plot of Pathway Analysis of ANCOVA (comparing metabolites after adjusting for age) results that was conducted by hypergeometric test using Homo sapiens KEGG library as the reference. The X-axis indicating the impact or importance of respective pathways on Mn exposure study. The Y-axis describes the statistical significance of results (*p*-value). The size of bubble shows the number of compounds involved in the pathway that are affected by the exposure. Larger bubbles mean a greater number of compounds. The bubble color represents the magnitude of *p*-value, a more intense red color indicates a lower *p*-value.

**Figure 8 brainsci-15-00829-f008:**
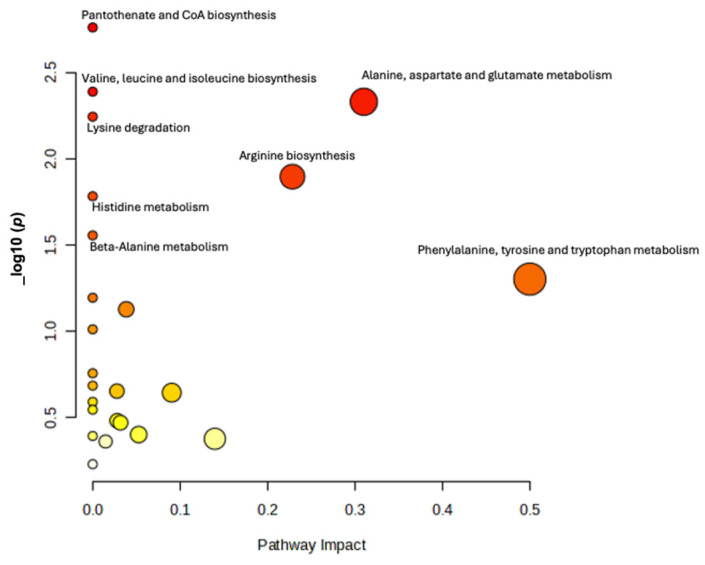
Scatter plot of Pathway Analysis of Linear regression (where metabolites were modeled as a dependent variable, plasma manganese as independent variable, and age as covariate) results that was conducted by hypergeometric test using Homo sapiens KEGG library as the reference. The X-axis indicating the impact or importance of respective pathways on Mn exposure study. The Y-axis describes the statistical significance of results (*p*-value). The size of bubble shows the number of compounds involved in the pathway that are affected by the exposure. Larger bubbles mean a greater number of compounds. The bubble color represents the magnitude of *p*-value, a more intense red color indicates a lower *p*-value.

**Figure 9 brainsci-15-00829-f009:**
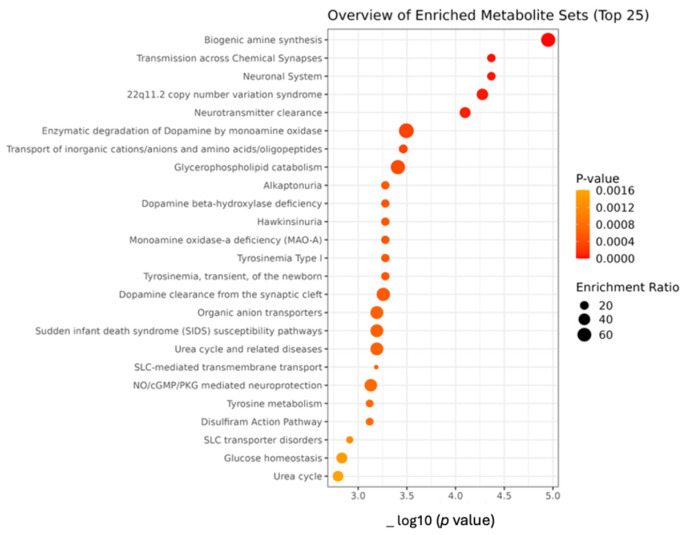
Over-representation analysis (ORA). It was used for enrichment assessment of ANCOVA (adjusted for age) results using RaMP-DB as the reference indicating the top 25 enriched metabolite set in the study of metabolite profile of Mn-exposed workers than controls. Dot plot showing the *p*-value on the X-axis and metabolites set on the Y-axis. The size of the dots reflects the enrichment ratio, indicating is a measure of the magnitude by which a set of metabolites is over-represented in the dataset compared to a background reference set. Larger dots indicate a higher enrichment ratio. The color represents the *p*-value, the color scale goes from yellow (less significant) to red (more significant).

**Figure 10 brainsci-15-00829-f010:**
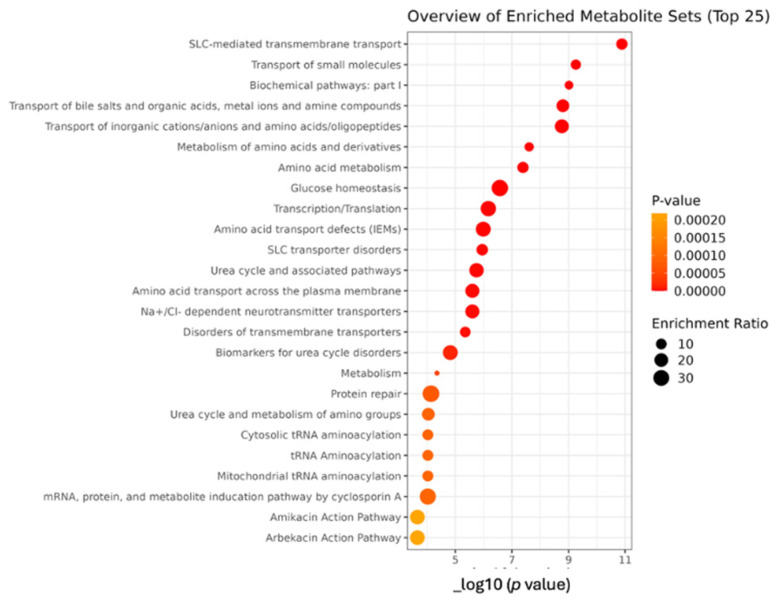
Dot plot of over-representation analysis (ORA). ORA used for enrichment assessment of linear regression results (where metabolites were modeled as a dependent variable, plasma manganese as independent variable, and age as covariate) using RaMP-DB as the reference indicating the top 25 enriched metabolite set in the study of metabolite profile of Mn-exposed workers than controls.

**Figure 11 brainsci-15-00829-f011:**
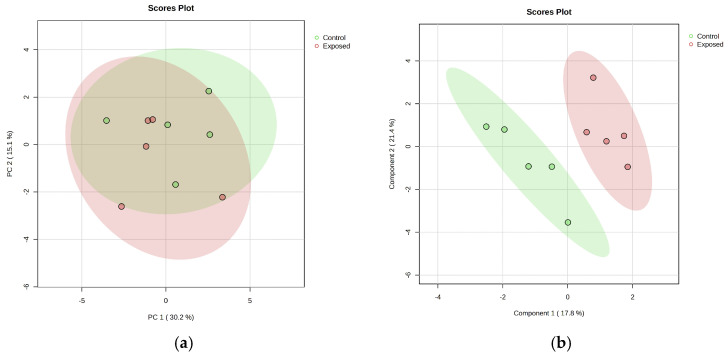
(**a**) Principal Component Analysis (PCA) of lipidomic data indicates a slightly distinct discrimination between Mn-exposed and control group. (**b**) Partial Least Squares Discriminant Analysis (PLS-DA) of metabolomic data shows a distinct discrimination of Mn-exposed and control groups.

**Figure 12 brainsci-15-00829-f012:**
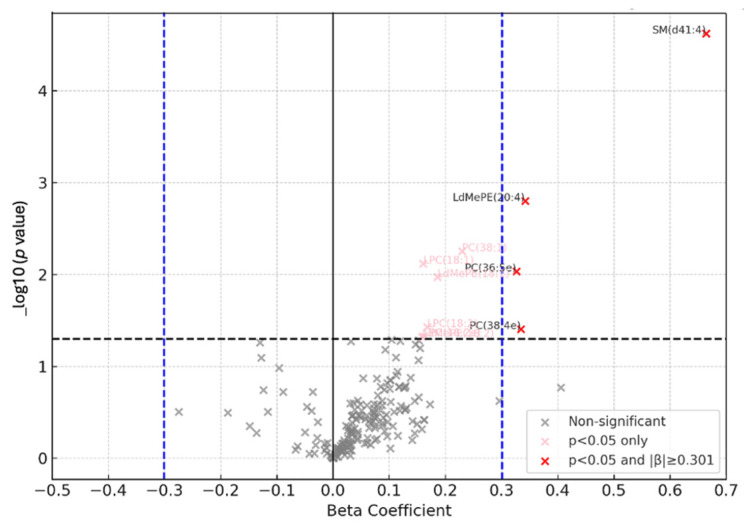
Volcano plot depicted the linear regression results (where plasma Mn was treated as independent variable, Lipid, as dependent variable, and Age as a covariate in the model) comparing lipids between the manganese-exposed group and controls controlling for age. Each point represents a metabolite, with the X-axis indicating the beta coefficient (the strength and directionality of change in metabolite level), and the Y-axis showing the p value. The thresholds for beta coefficient representing a 2-fold change (β = 0.301), and *p* < 0.05.

**Figure 13 brainsci-15-00829-f013:**
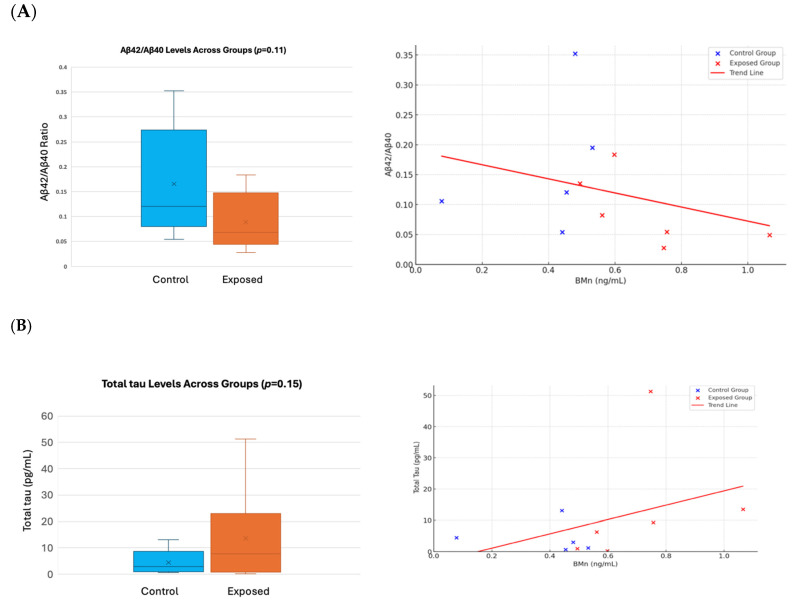
Comparative levels of the Aβ42/Aβ40 ratio and Total tau were measured using single-molecule arrays (Simoa) in Quanterix SR-X platform. (**A**) Left: Bar chart indicates the level of Aβ42/Aβ40 ratio in the plasma of Mn-exposed workers (Mean ± SD: 0.09 ± 0.06) and control group (Mean ± SD: 0.16 ± 0.1). Right: The Scatter plots represent the individual Aβ42/Aβ40 ratio values versus plasma Mn values. Blue points represent control individuals, while red points represent Mn-exposed individuals. The red line is a regression line indicating a trend or relationship between the plasma Mn and Aβ42/Aβ40 ratio across all samples. (**B**) Left: Bar chart indicates the level of Total tau in the plasma of Mn-exposed workers (Mean ± SD: 13.6 ± 19.1) and control group (Mean ± SD: 4.45 ± 5.04). Right: The Scatter plots represent the individual data points for proteins level versus plasma Mn values. Blue points represent control individuals, while red points represent Mn-exposed individuals. The red line is a regression line indicating a trend or relationship between the plasma Mn and Total tau protein concentrations across all samples.

**Figure 14 brainsci-15-00829-f014:**
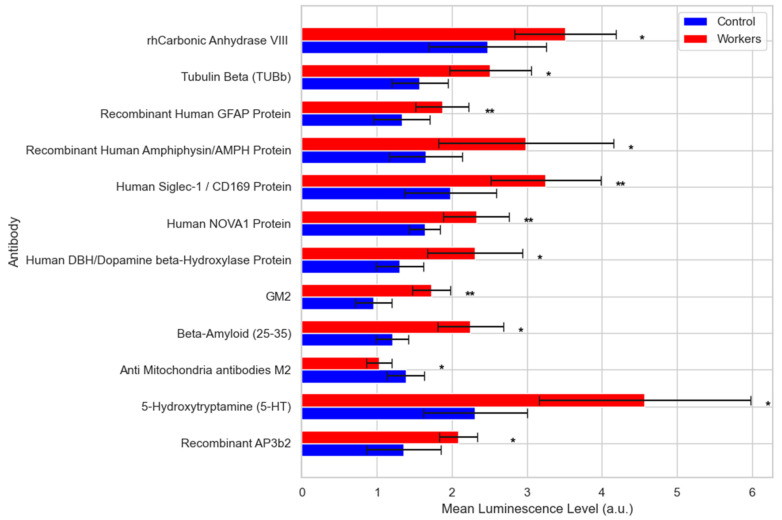
Bar chart depicting mean luminescence levels (±SD), reflecting relative plasma antibody abundance targeting autoimmune and neurological proteins in manganese (Mn)-exposed workers and unexposed controls. Antibody signals were quantified using a protein microarray platform. Statistical significance was determined using unpaired two-tailed Welch’s *t*-tests. Significance is indicated as follows: *p* < 0.05 (*), *p* < 0.001 (**).

**Figure 15 brainsci-15-00829-f015:**
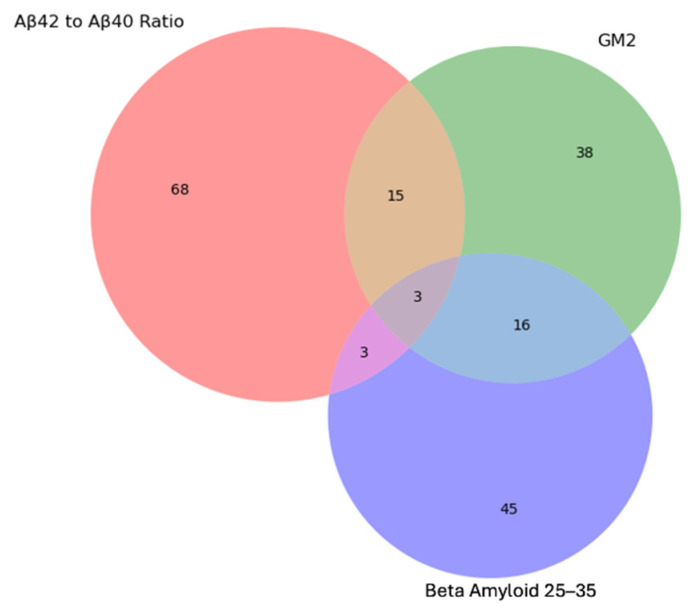
Venn diagram showing shared significant (*p* < 0.05) metabolites associated with β-amyloid (25–35), GM2, and the Aβ42/Aβ40 ratio. The diagram was generated in Python to illustrate overlapping metabolic signatures across these three neurodegenerative markers.

**Table 1 brainsci-15-00829-t001:** Demographic and Descriptive Statistics table of Study Population.

Characteristics	Exposed N (%)	Controls N (%)	Statistic	*p* Value
Age (years)				
54–61	3 (50%)	3 (60%)	Chi2 = 0.0	1.00
62–69	3 (50%)	2 (40%)		
Smoking Consumption				
YES	3 (50%)	3 (60%)	Chi2 = 0.0	1.00
NO	3 (50%)	2 (40%)		
Alcohol Consumption				
YES	5 (83%)	3 (60%)	Chi2 = 0.03	0.85
NO	1 (17%)	2 (40%)		
Comorbidity				
YES	5 (83%).	4 (80%)	Chi2 = 0.0	1.00
NO	1 (17%)	2 (20%)		
ND Family History *				
YES	3 (50%)	2 (40%)	Chi2 = 0.0	1.00
NO	3 (50%)	3 (60%)		

* ND: Neurodegenerative Diseases.

**Table 2 brainsci-15-00829-t002:** Pathway analysis using lipidomic data of linear regression model. It is showing significant impacted pathways in comparison of Mn-exposed groups rather than controls. The significantly different lipids of Linear regression results used for lipids pathway analysis using LIPEA, Homo sapiens lipids pathways used as reference for background. PMn was treated as independent variable, Lipid, as dependent variable, and Age as a covariate in the model.

Pathway Name	Pathway Lipids	*p*-Value	Benjamini Correction
Glycerophospholipid metabolism	26	0.00039213	0.0018621
Choline metabolism in cancer Necroptosis	5 4	0.0004138 0.02954564	0.0018621 0.0886369
Retrograde endocannabinoid signaling	8	0.05843168	0.1179923

## Data Availability

The original contributions presented in this study are included in the article/Appendix A. Further inquiries can be directed to the corresponding author.

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
