# Peer review of "Impact of Manganese on Neuronal Function: An Exploratory Multi-Omics Study on Ferroalloy Workers in Brescia, Italy"

_brainsci, 2025, doi:10.3390/brainsci15080829_

Round 1
Reviewer 1 Report
Comments and Suggestions for Authors
The manuscript by Somaiyeh Azmoun and colleagues is a pilot study assesing the impact of chronic manganese intoxication on brain function and neurodegeneration. The authors identified many novel metabolic and immunologic markers of manganese-mediated brain damage. The data obtained allow not only to propose new markers for the diagnosis of manganese-mediated brain damage, but also to suggest possible or confirm the known molecular and cellular mechanisms of such damage. The work is performed at a high experimental level. Despite the advantages of this work, it is necessary for the authors to correct some errors and answer some questions:
1. what specific molecular markers allows one to conclude that olfactory signaling is disrupted? how can metabolites testify about damage of certain brain neural pathways? no such information was found in the manuscript or supplementary materials. is it specific or another interpretation of these data is possible?
2. Lines 516-518 - there is a mistake: Citric acid, Hypoxanthine, Lactic acid, Uracil, Creatinine, N-Acetylneuraminic acid are NOT amino acids or oligopeptides
3. how do the authors explain: "Except for anti-mitochondrial an-631 tibodies M2, which were decreased" ? when tissue is disrupted one should observe the opposite changes - increased mitochondrial release and increase titer of antibodies to mitochondria.
4. Lines 611-616 are redundant - the authors list factors and mechanisms that are not investigated or discussed in the paper (at the authors' discretion)
5. Line 628 remail - is it OK?
6. homosapiens - is it OK?
7. figure 14: the vertical axis is named "mean abs level" - why not "mean luminescence level" if luminescence was measured? also the st.dev. spreads for each bar should be indicated or is it a result of a single measurement?
8. not all figures contain an indication of the number of measurements. it should be supplemented.
Reviewer 2 Report
Comments and Suggestions for Authors
- In the introduction, the connection between Manganese neurotoxicity and Alzheimer’s Disease is somewhat unclear. Consider adding more research to support these links: expanding on how Manganese deteriorates the brain function, more pathway could enhance the reader's understanding of its specific effects within these mechanisms.
- Please provide a Clinical Trial Number in the manuscript.
- Briefly specify the statistical tests used to determine significance.
- Add future directions to the discussion section.
Some grammatical improvements could enhance clarity, particularly in the Abstract and Introduction.
Round 2
Reviewer 1 Report
Comments and Suggestions for Authors
The authors have made all necessary corrections and replied to all the questions posed.
I recommend to accept the manuscript in the present form.